# TAPES: A tool for assessment and prioritisation in exome studies

**Alexandre Xavier**[1]*, **Rodney J. Scott**[1,2], **Bente A. Talseth-Palmer**[1,3]

**1** School of Biomedical Sciences and Pharmacy, Faculty of Health and Medicine, University of Newcastle and Hunter Medical Research Institute, Newcastle, Australia, **2** NSW Health Pathology North, John Hunter Hospital, Newcastle, Australia, **3** Clinic for Research, Innovation, Education and Development, Møre and Romsdal Hospital Trust, Molde, Norway

* alexandre.xavier@live.fr

## Abstract

Next-generation sequencing continues to grow in importance for researchers. Exome sequencing became a widespread tool to further study the genomic basis of Mendelian diseases. In an effort to identify pathogenic variants, reject benign variants and better predict variant effects in downstream analysis, the American College of Medical Genetics (ACMG) published a set of criteria in 2015. While there are multiple publicly available software's available to assign the ACMG criteria, most of them do not take into account multi-sample variant calling formats. Here we present a tool for assessment and prioritisation in exome studies (TAPES, https://github.com/a-xavier/tapes), an open-source tool designed for small-scale exome studies. TAPES can quickly assign ACMG criteria using ANNOVAR or VEP annotated files and implements a model to transform the categorical ACMG criteria into a continuous probability, allowing for a more accurate classification of pathogenicity or benignity of variants. In addition, TAPES can work with cohorts sharing a common phenotype by utilising a simple enrichment analysis, requiring no controls as an input as well as providing powerful filtering and reporting options. Finally, benchmarks showed that TAPES outperforms available tools to detect both pathogenic and benign variants, while also integrating the identification of enriched variants in study cohorts compared to the general population, making it an ideal tool to evaluate a smaller cohort before using bigger scale studies.

## Author summary

New sequencing techniques allow researchers to study the genetic basis of diseases. Predicting the effect of genetic variants is critical to understand the mechanisms underlying disease. Available software can predict how pathogenic a variant is, but do not take into account the abundance of a variants in a cohort. TAPES is a simple open-source tool that can both more accurately predict pathogenicity (using probability over categories) and provide insight on variants enrichment in a cohort sharing the same disease.

**Data Availability Statement:** All source code can be found at: https://github.com/a-xavier/tapes. Documentation is also available through this repository and at: https://github.com/a-xavier/tapes/wiki.

**Funding:** The Hunter Cancer Research Alliance (https://www.hcra.com.au/) funded Bente Talseth-Palmer and Alexandre Xavier. The University of Newcastle (https://www.newcastle.edu.au/) funded Alexandre Xavier. The Cancer Institute NSW (https://www.cancercouncil.com.au/) funded Bente Talseth-Palmer. The funders had no role in study design, data collection and analysis, decision to publish, or preparation of the manuscript.

**Competing interests:** The authors have declared that no competing interests exist.

This is a *PLOS Computational Biology* Software paper.

## Introduction

With the advances in Next-Generation Sequencing (NGS) technologies and the decline in price over the last few years, exome sequencing has become a standard tool to explore the genetic basis of inherited diseases [1]. It has become easy to annotate the ever-increasing amount of variants identified by such methods, using tools such as VEP [2], snpEff [3] or ANNOVAR [4]. These tools help researchers to better predict the downstream effect of a variant and give insight, for example, on the frequency of the mutation in the general population, the impact on proteins or in-silico predictions of pathogenicity.

In 2015, the American College of Medical Genetics (ACMG) published a set of criteria to assess the probability of a variant pathogenicity, classifying them into five categories [5], from benign to pathogenic, facilitating downstream analysis.

Since then, tools have been developed to assess individual variant pathogenicity using the ACMG criteria (such as CharGer [6] and Intervar [7]) but they do not have the ability to take into account the frequency of variants in a cohort. The categorical nature of the ACMG criteria also leaves a lot of variants classified as "a variant of unknown significance".

Here, we present TAPES, an open-source tool to both assess and prioritise variants by pathogenicity. TAPES can assign the ACMG criteria and by using one of the first implementations of the model described in Tavtigian *et al.* [8], providing a more nuanced and easy to understand estimated probability for a variant to be either pathogenic or benign, thus transforming categorical classification into a more linear prediction. Our goal during development was first to create a simple tool that can better predict pathogenicity and reject benign variants, and then to assess a cohort sharing a phenotype by detecting enriched variants compared to the general population without the need of control samples. In addition, we focused on providing simple yet powerful reporting and filtering systems, while allowing pathway analysis of pathogenic mutations, gene-burden calculations and per-sample reporting.

## Design and implementation

### ANNOVAR interface and annotated variant file

TAPES sorting option can be used with both ANNOVAR and VEP annotated variant calling files (VCF). However we also provide users with simple wrapping tools for a local installation of ANNOVAR to simplify the workflow (this requires users to download ANNOVAR). Users can annotate VCF, gzipped VCF and binary VCF (BCF) using two simple commands without having to specify the databases and annotations to use.

While there are a set of annotation needed to assign all ACMG criteria (see https://github.com/a-xavier/tapes/wiki/Necessary-Annotations for the full list), TAPES will use as many available annotations as possible to assign the relevant ACMG criteria.

### Variant classification

**TAPES** requires annotated ANNOVAR (VCF or tab/comma-separated values) or VEP (VCF) files to use the sorting module.

**Regular ACMG criteria assignment.**   For most of the ACMG criteria assignment (PVS1, PS1, PS3, PM1, PM2, PM4, PM5, PP2, PP3, PP5, BS1, BS2, BS3, BP1, BP3, BP4, BP6, BP7 and BA1), we tried to stay as true to the original ACMG definition as possible when implementing

the criteria assignment. Please see Richards *et al*. [5] and S1 Table for more information on the ACMG Criteria definition.

**Enrichment analysis / PS4 criteria.**   One of TAPES unique features is the ability to calculate variant enrichment from public frequency data (ExAC or gNomad [9]), without having to sequence control samples. In cohort studies, TAPES require a multi-sample vcf file to extract genotyping data and get frequencies from the cohort studied. It uses a simple one-sided Fisher's exact test to calculate both the Odds Ratio (OR) and the p-value of the enrichment. Only the variant enrichment in the cohort is tested against the general population.

Since OR calculation requires integer numbers and frequency in the general population is given as a 0–1 fraction, TAPES approximates the number of individuals affected using the following formula.

If $MAF_c$ is the Minor Allele Frequency (MAF) in a control population, $n_c$ is the number of individuals affected by the variant in the control population and $N_c$ is the number of individuals without the variant then:

$$MAF_c = y \times 10^{-x}, n_c = \lceil y \rceil \text{ and } N_c = \frac{10^x}{2} - n_c.$$

For example if:

$$MAF_c = 3.23 \times 10^{-5} \text{ then } n_c = 4 \text{ and } N_c = \frac{10^5}{2} - 4.$$

This approximation is only valid if the following assumptions are made; MAF in the control population is under 0.05 and that very rare variants are mostly heterozygous.

The PS4 criteria assignment was designed to be more stringent than a normal study with controls (choosing to overestimate the frequency in the general population) and will only be assigned if OR $\geq$ 20, p-value $\leq$ 0.001 and at least 2 individuals in the cohort share the variant.

**Trio analysis / PS2 assignment.**   TAPES allow researchers to work with trio studies. In trio studies, the user provides information such as sample name, trio ID and pedigree information in a tab-delimited file. Then PS2 will be assigned if a variant is identified as *de-novo* and healthy parents are removed from downstream analysis. PS2 is assigned to a variant if it was found as *de-novo* in any trio but details from each trio will still be provided.

**Probability of pathogenicity calculation.**   TAPES includes the model developed by Tavtigian *et al*. [8] to transform ACMG categorical classification into linear probability of pathogenicity and the method uses the default parameters from (Prior P = 0.10, $O_{PVSt}$ = 350 and X = 2). This allows for a finer pathogenicity prediction and adjustable thresholds to decide variant pathogenicity. It is important to keep in mind that this measure is a probability and not a measure of how pathogenic a variant is.

## Cohort reporting

**TAPES** provides an array of different useful reports.

**Filtering.**   TAPES can easily perform advance filtering. Three different options are available. First, users can provide a custom list of gene symbols (either as a text file or directly on the command line) to only output variants present in those genes. Then users can also do a reverse pathway search by providing the name of a pathway (extracted from KEGG pathways [10]) and output a report with variants in genes involved in that pathway. Finally, users can run searches based on terms contained in the description for each gene, i.e. if the user looks for 'autosomal dominant' genes or 'colorectal cancer' genes. These filtered reports keep the same format as the main report, making it possible to use them with other reporting tools.

**By-sample report.** For each individual in the cohort, a report containing the variant predicted to be pathogenic with the highest level of confidence will be available. This allows the study of individual samples and their specificity.

**By-gene report.** TAPES can also calculate, for each gene, a gene burden score. This score helps determining which genes harbour the most potentially pathogenic variants in a cohort. This can be useful when searching for variants in diseases caused by single genes and that cannot be discovered using pathway analysis. The gene burden score is calculated by summing the probability of pathogenicity of a specific variant multiplied by the number of individuals with that genotype in the cohort.

$$Gene\ burden\ score = \sum_1^n P_i \times N_i$$

Calculated for each gene, where $P_i$ = the probability of pathogenicity of the variant and $N_i$ = Number of samples affected by the variant. If $P_i \leq 0.80$ then the variant is excluded.

This measure is useful to detect which genes in the cohort are particularly enriched in pathogenic and probably pathogenic variants (it is important to remember that this measure is a sum of probabilities). However, there are a few caveats. This measure might be affected by very long genes or genes frequently mutated in exomes (FLAGS [11]). In some cases, poorly mapped reads (for example due to pseudo-autosomal regions in the X or Y chromosome), might impact the result with an excessive number of samples affected by a variant. TAPES provides an appropriate warning for all of those cases.

**Pathway analysis.** TAPES can also perform a pathway analysis using the EnrichR [12] API. Only genes containing variants that are predicted to be pathogenic are kept as a gene list. The user can then use any library to analyse the gene list but the default is GO_Biological_Process_2018. Pathway analysis is important to understand the possibly disrupted mechanism and the commonalities between variants found in a cohort.

## Results

### Variant classification

TAPES variant classification was benchmarked against similar tools, CharGer [6] and Intervar [7] using the prediction on the pathogenicity of variants of the expert panel of Zhang et al., 2015 as reference [13] (see S2 Table for the full table). This dataset was also used to benchmark CharGer in their original publication. The 'probably pathogenic' and 'pathogenic' predictions were pooled into one 'pathogenic' group. Similarly the 'probably benign' and 'benign' were pooled into one 'benign' group.

To assess the predictive power of each software, we used Receiver Operating Characteristics (ROC) curves and calculated the area under the curve (AUC) as well as the precision-recall curves and average precision (AP). We compared TAPES ACMG and probability of pathogenicity prediction with CharGer score and InterVar ACMG prediction (see Fig 1).

TAPES probability of pathogenicity, using Tavtigian *et al* [8] modelling, outperformed both software's tested using AUC and AP for prediction of both pathogenic and benign variants.

AUC and AP show that using TAPES ACMG criteria assignment remains less precise than using CharGer custom score (due to the additional information CharGer need to function properly) and closer to InterVar. Using the probability of pathogenicity should be the preferred way to identify pathogenic variants and reject benign variants. Based on ROC curves, a threshold of 0.80–0.85 for probability of pathogenicity seemed to keep high true positive rate (TPR) while low false positive rate (FPR) for predicting pathogenic variants. Similarly, a

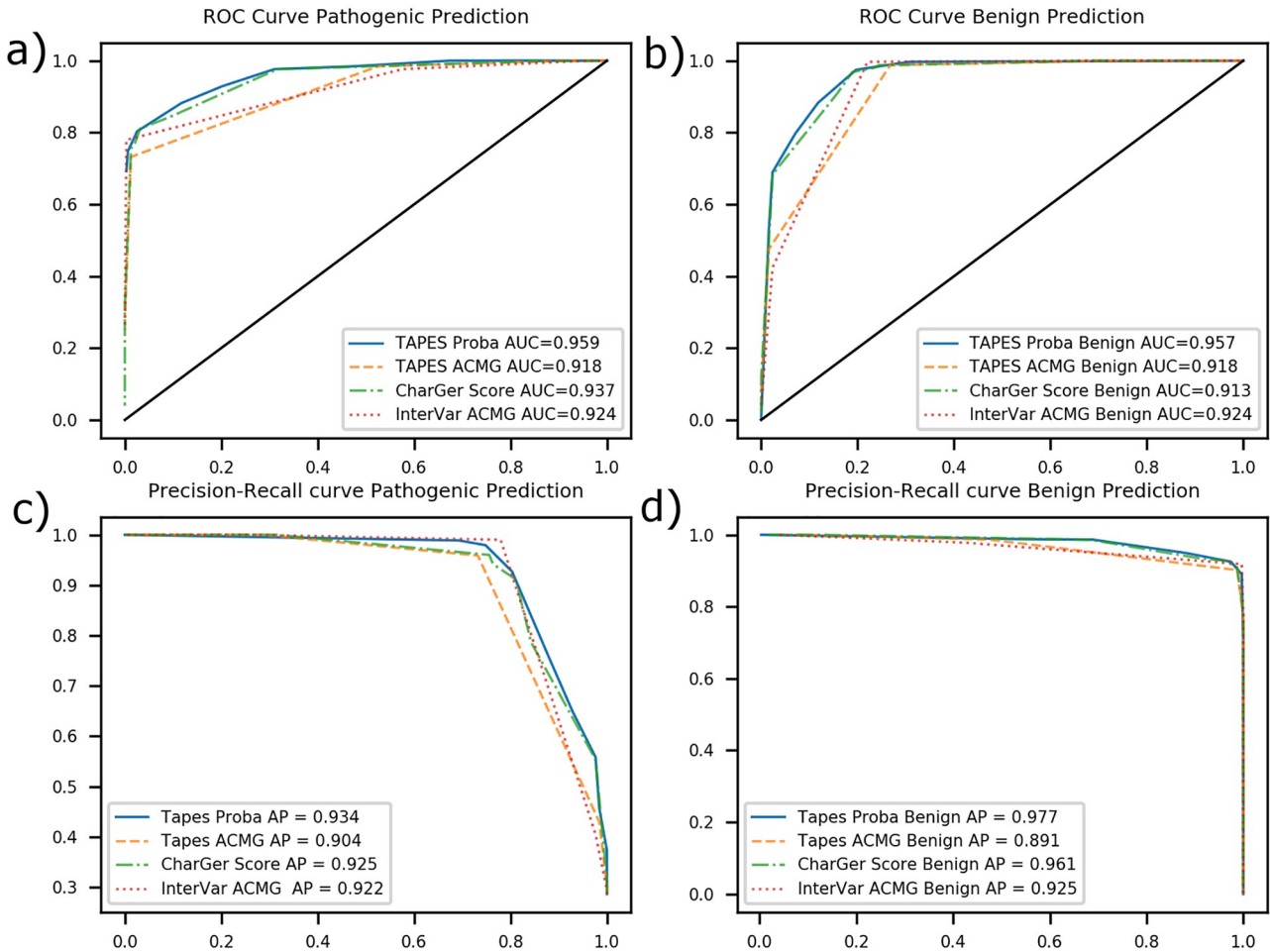

**Fig 1. ROC curves and precision recall curves.** a) ROC curve of various softwares for pathogenicity prediction AUC b) ROC curve of various softwares for benignity prediction AUC c) Precision-recall curve of various softwares for pathogenicity prediction d) Precision-recall curve of various softwares for benignity prediction (Metrics used; TAPES proba; TAPES probability of pathogenicity prediction, TAPES ACMG: TAPES ACMG prediction, CharGer score: CharGer prediction of pathogenicity based of a custom score, InterVar ACMG: InterVar ACMG prediction).

threshold of 0.20–0.35 for probability of pathogenicity had high TPR and low FPR for predicting benignity.

To validate these findings and choose the best probability thresholds for pathogenicity and benignity, we used TAPES, InterVar and CharGer on a different dataset **(see** S3 Table**)**. Using 530 hand curated variants from ClinGen evidence repository (https://erepo.clinicalgenome.org/evrepo/) as ground truth. TAPES outperformed both InterVar and CharGer **(see** Fig 2**)**. In addition to the precision of the prediction, TAPES also outperformed other software in terms of absolute number of variants correctly identified.

We recommend to use TAPES probability of pathogenicity prediction with either lenient thresholds of 0.8 and 0.35 (respectively for pathogenicity and benignity) or stricter thresholds of 0.85 and 0.20.

## Variant enrichment / PS4 benchmark

We compared our method of calculation of ORs compared to the normal method (see Fig 3).

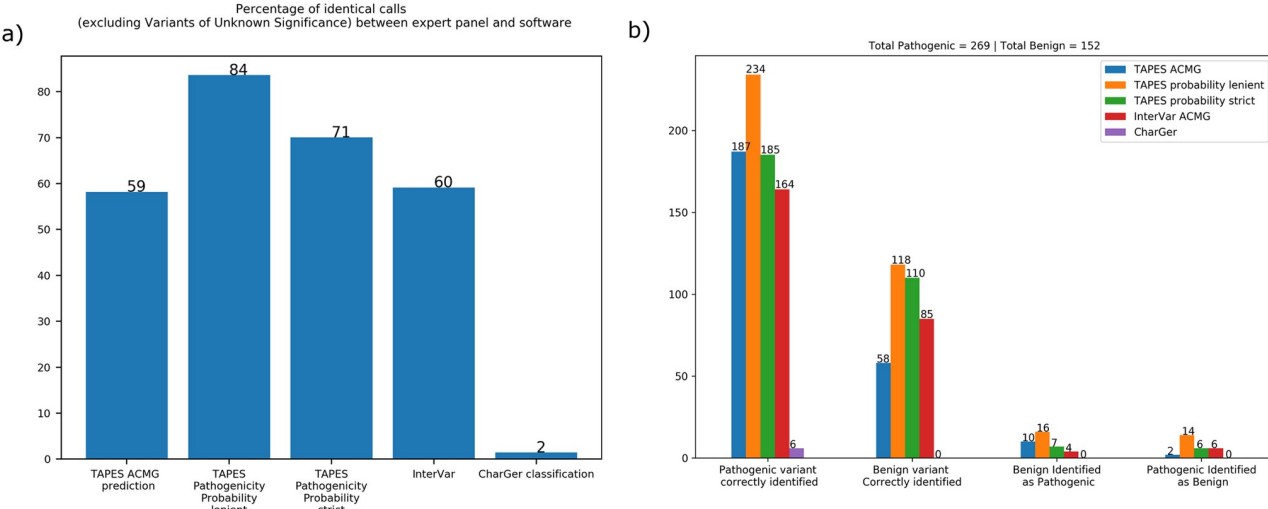

**Fig 2. Validation dataset software comparisons. a) Percentage of identical calls between the ClinGen expert panel decisions and software prediction**. Lenient thresholds are 0.80 for pathogenicity and 0.35 for benignity. Strict thresholds are 0.85 for pathogenicity and 0.20 for benignity. **b) Absolute number of variants predictions**. Pathogenic and benign variants correctly and incorrectly identified between the panel of expert and various software. (Metrics used; TAPES probability lenient; TAPES probability of pathogenicity prediction 0.35–0.80, TAPES probability strict; TAPES probability of pathogenicity prediction 0.20–0.85 TAPES ACMG: TAPES ACMG prediction, CharGer: CharGer prediction of pathogenicity based of a custom score, InterVar ACMG: InterVar ACMG prediction).

The OR using TAPES extrapolation is always smaller than the normal calculation, making it more stringent. Similarly, the p-value of the Fisher's exact test rises faster with frequency than the normal method. This way, only the most significantly enriched variant are assigned with PS4 to ensure very few false positives.

**Reporting options.** TAPES reporting options are powerful and easy to use. Using a mock input file with variants from Zhang *et al.* [13] as well as simulated samples to form a cohort, the pathway analysis correctly identified DNA repair as the pathway with the most probable pathogenic variants.

The by-gene report also identified BRCA2 as the gene with the highest gene burden.

See S1 File to see all reports templates.

## Availability and future directions

TAPES is available on github at: https://github.com/a-xavier/tapes, under the MIT licence, which allows anyone to both freely download and modify the source code. Help can be found both in the manual (located in the main repository) or on the wiki (https://github.com/a-xavier/tapes/wiki). Examples of inputs can also be found in the main repository. Dependencies can be easily installed using PyPi repositories (pip). All builds are verified through Travis continuous integration on Linux, Windows and macOS. All benchmarks and examples showed in this manuscript were generated using TAPES release 0.1.

All benchmarks and examples were generated using the initial release 0.1 of TAPES (https://github.com/a-xavier/tapes/releases).

TAPES will continue to evolve with the advances in various databases such as ExAC, dnSNP or dbNSFP. As they constantly update their data and the format, TAPES will evolve to be more precise and accurate. In addition, future directions include more statistical measures to detect significant variants in different cohort studies.

PS4 calculation with Fisher's exact test one-sided (greater)

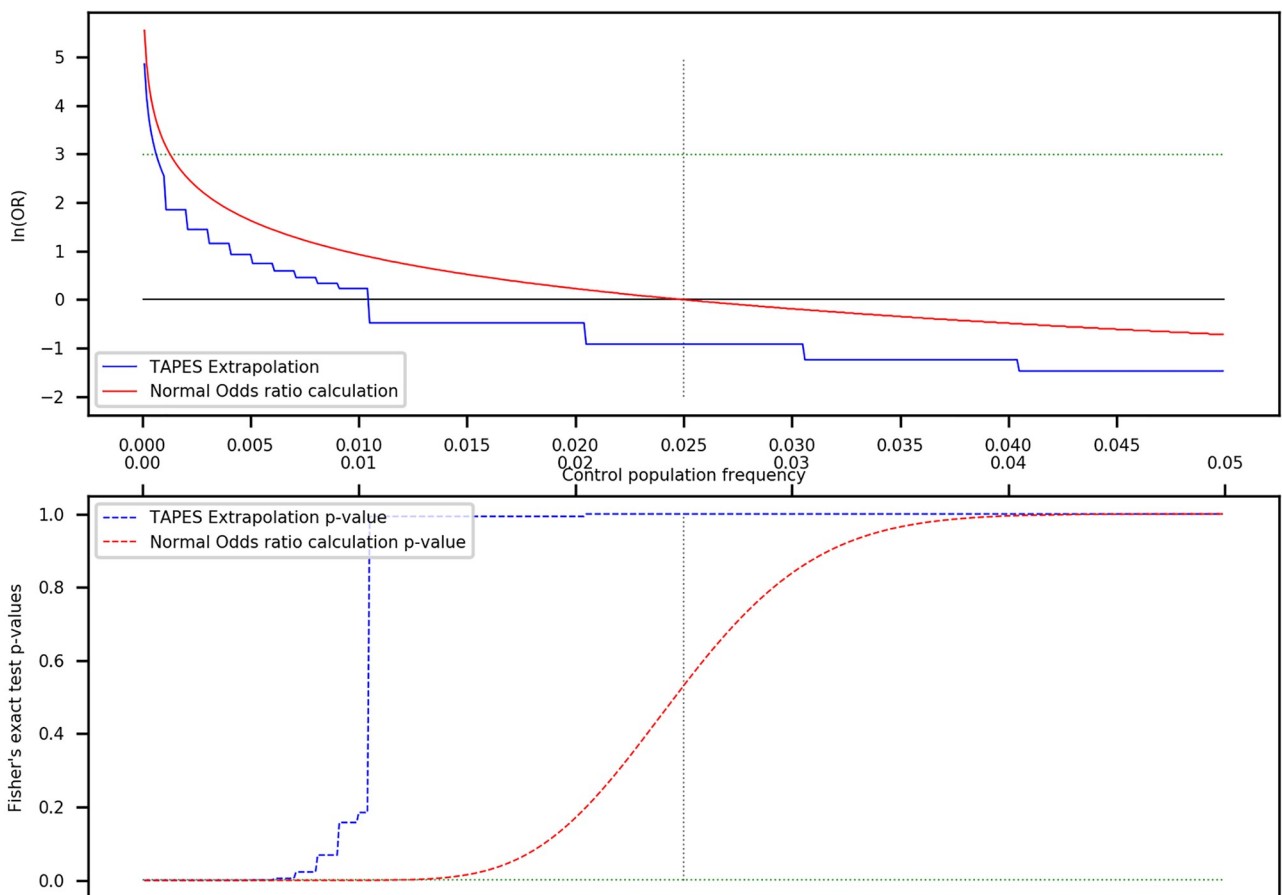

**Fig 3. PS4 calculation with Fisher's exact test one sided.** Comparison of TAPES extrapolation of odds rations compared to the normal method (top graph). Comparison of the p-value of both methods (bottom graph). The **vertical dotted line** represents the known frequency of the variant in the studied cohort (0.025). The **horizontal green dotted line** represents the thresholds used to assign PS4 (OR = 20 or ln(OR) = 2.9957 (top) and p-value < 0.01(bottom)).

We aim to keep TAPES as simple and useful as possible to make it a perfect endpoint tool to analyse variants from small-scale cohorts.

## Supporting information

**S1 Table. ACMG criteria assignment in TAPES and definitions from the original Richards et al 2015 article.**
(XLSX)

**S2 Table. Comparison of Prediction between different pathogenicity assessment software and the expert panel from Zhang J et al. 2015.** Comparison between TAPES ACMG and pathogenicity probability prediction, CharGer Prediction Score and InterVar AMCG Prediction.
(XLSX)

**S3 Table. Comparison of Prediction between different pathogenicity assessment software and the expert panel from ClinGen evidence repository variants.** Comparison between TAPES ACMG and pathogenicity probability prediction, CharGer Prediction Score and Inter-Var AMCG Prediction.
(TXT)

**S1 File. Example reports from TAPES sort option.** Generated using the data from: Zhang, J., et al. Germline Mutations in Predisposition Genes in Pediatric Cancer. N Engl J Med 2015;373(24):2336–2346. Using the command: *python tapes.py sort -i ./input.csv -o ./Report/ - -tab - -by_gene - -by_sample - -enrichr - -disease "autosomal dominant" - -kegg "Pathways in cancer"*. This file gives examples for the main report, the by-gene report, the by-sample report, the enrichr report, the disease report and the kegg report.
(XLSX)

**S2 File. Files used for TAPES benchmark and validation. The Initial Benchmark folder contains all files used for the original benchmark, CharGer_and_Panel_Benchmark.xlsx:** CharGer pathogenicity prediction and expert panel decision from from: Zhang, J., et al. 2015, extracted from CharGer original publication, **Synthetic_VCF_for_Benchmark.vcf.vcf:** Synthetic VCF file created from the **CharGer_and_Panel_Benchmark.xlsx** variants information, **InterVar_Benchmark.txt:** InterVar predictions of pathogenicity after analysis of the synthetic VCF, **TAPES_Benchmark.xlsx**: TAPES prediction of pathogenicity after analysis of the synthetic VCF. The results of all 3 software are compiled in S2 Table. **The Validation folder contains all filed used for the validation of the pathogenicity thresholds and comparison with other software. TAPES_validation_synthetic.vcf**: Synthetic VCF created with data extracted from the ClinGen evidence repository (https://erepo.clinicalgenome.org/evrepo/), **TAPES_validation.charger.txt:** the CharGer predictions of pathogenicity after analysis of the Synthetic VCF, **TAPES_Validation.intervar.txt:** InterVar prediction of pathogenicity after analysis of the synthetic VCF, **TAPES_Validation.tapes.txt:** TAPES prediction of pathogenicity after analysis of the Synthetic VCF. The results of all 3 software are compiled in S3 Table.
(ZIP)

## Acknowledgments

The authors would like to thank Mr. Sean Burnard for his helpful advices regarding this manuscript.

## Author Contributions

**Conceptualization:** Alexandre Xavier.

**Formal analysis:** Alexandre Xavier.

**Funding acquisition:** Rodney J. Scott, Bente A. Talseth-Palmer.

**Methodology:** Alexandre Xavier.

**Resources:** Rodney J. Scott.

**Software:** Alexandre Xavier.

**Supervision:** Bente A. Talseth-Palmer.

**Writing – original draft:** Alexandre Xavier.

**Writing – review & editing:** Rodney J. Scott, Bente A. Talseth-Palmer.

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
