## [Decision Letter · Decision Letter 0]

2 Aug 2019

Dear Dr Xavier,

Thank you very much for submitting your manuscript 'TAPES: a tool for assessment and prioritisation in exome studies' for review by PLOS Computational Biology. Your manuscript has been fully evaluated by the PLOS Computational Biology editorial team and in this case also by independent peer reviewers. The reviewers appreciated the attention to an important problem, but raised some substantial concerns about the manuscript as it currently stands. While your manuscript cannot be accepted in its present form, we are willing to consider a revised version in which the issues raised by the reviewers have been adequately addressed. We cannot, of course, promise publication at that time.

Sincerely,

Mihaela Pertea

Software Editor

PLOS Computational Biology

Mihaela Pertea

Software Editor

PLOS Computational Biology

[LINK]

Reviewer's Responses to Questions

**Comments to the Authors:**

Reviewer #1: SUMMARY:

'TAPES: a tool for assessment and prioritisation in exome studies' implements a novel and more precise method for assessing variant pathogenicity by introducing a novel modeling for integration of ACMG criteria. They leverage this model along with publicly available variant population frequencies to provide more accurate predictions of variant pathogenicity. Additionally, this software provides a comprehensive list of both reporting and analysis options.

MAJOR CODE PROBLEMS:

- Code doesn't seem to have any tests or automated way to run them: https://github.com/a-xavier/tapes. Please add tests (preferrably using a testing framework such as PyTest) that minimally take advantage of your toy datasets that covers most of your functionality. Integration with a free and automated continuous integration environment like Travis would also be highly recommended. Once tests are in place, potentialy using branches to provide a more stable development path may aid development

- Toy example provided doesn't work natively or within a virtualenv:

- python3 tapes.py sort -i ./Example_Output/input.csv -o ./Toy_dataset/ --tab --by_gene --by_sample --enrichr --disease "autosomal dominant" --kegg "Pathways in cancer":

No acmg_db path given and no db_config.json found

Default is: /home/ubuntu/repositories/tapes/acmg_db

***TAPES: SORT***

2019-07-15 10:37:05.....Output type: FOLDER

Traceback (most recent call last):

File "tapes.py", line 309, in <module>

main()

File "tapes.py", line 164, in main

output_prefix = args.output.split('\\\\')[-2]

IndexError: list index out of range

MINOR CODE FEEDBACK

- I would put code that is not top-level in a `src` or `source` directory.

- While the Manual is fine as a PDF, long-term maintenance might be easier if it is in markdown. It can be further extrapolated using something like Read the Docs or other services.

MINOR EDITS:

- line 25: should be: "share the same phenotype" , missing "the"

- line 27: I think it reads better to say "Benchmarks showed that TAPES outperforms avaialable tools"

- line 34: "Available software can predict" drop "'s"

- line 90: "individuals affected" and "number of individuals", individuals I believe should be plural in both cases

- line 96: "very vare variants" I believe variants should be plural

- line 134: "cohort are in the class are probably" , missing "are"

- line 137-139: This is not a complete sentence.

- Figure 1: Charger should be "CharGer" in your legend

MINOR QUESTIONS:

- In this model there are no controls, which is novel. I'm mildly curious if it can be shown that providing controls offers little or no statistical benefit over the publily available variant frequencies.

- A minor discussion of why the CharGer Scores were so simlar to the TAPES probability model might be useful in context of Figure 1.

Reviewer #2: The article "TAPES: a tool for assessment and prioritisation in exome

studies" describes a new software tool to identify pathogenic and

benign variants. The aim described is very promising. However, I think

the clarity of both the paper and the documentation could be improved.

I will first comment on my experience with the software and then on

the paper. (This review is writen in MarkDown format, so it can be

converted to html or other format to see code section.)

## Comments on the software package.

I cloned the repository from GitHub, and could install it. Then I ran

into a few problems.

1. I found a bug in `t_func.py` on line 3197 that made the program

crash

```

with gzip.open(os.path.join(acmg_db_path, 'repeat_dict.{}.gz'.format(assembly)), "r") as dj:

```

Correcting the line to the following solved the issue.

```

with gzip.open(os.path.join(acmg_db_path, 'repeat_dict.{}.gz'.format(assembly)), "rt") as dj:

```

2. It was not clear at installation that I should install annovar to

be able to use tapes. I have found this information later in the

manual.

3. After installing annovar I needed to run

`python3 tapes.py db -s -A annovar` and

`python3 tapes.py db -b annovar` before I could annotate vcf

files. These commands were only mentioned at the end of the manual.

4. I did not manage to find a way to start with a vcf file, annotate

it and finally obtain ACMG classification. I think a tutorial and an

example dataset (starting from vcf files) would be valuable additions.

5. I would suggest to add a workflow diagram both to the manual and to

the paper to make it clear what kind of steps are needed and what

are the potential input and output files.

6. I could not identify what was the input file used for the analysis

shown in the paper, so I could not check whether it is

reproducible.

7. The program does not always produce the expected file name or it

does create the expected file, but does not log it

correctly. I think the code needs to be checked more thoroughly.

8. Please create a release for the publication version of the package

so people can know which version/status of the software was used

for the publication (This can be done at https://github.com/a-xavier/tapes/releases).

9. A docker image is always a nice addition, to make sure that

everything is specified as it should be, and there are no problems

due to difference in the software environment. It is also a good

way to test, how a software can be installed in a new environment.

## Comments on the paper

I have found several typos and grammatically mistakes. I think the

text should be checked more thoroughly for mistakes.

1. Abstract line 17: What does "downstream" variants refer to?

2. On line 20 multi-sample variant calling formats are mentioned in

the abstract, but this is never mentioned further in the article. I

would either remove it from the abstract or add an explanation to

a later section.

3. Lines 25-26. The Authors mention that cohort samples can be analyzed

even without a control sample set. My question is whether it is

possible to make use of a control set or is it only possible to use

the standard option where the databases are checked?

4. Lines 26-27: "Finally, it can provide powerful filtering and

reporting options to help researchers make sense of cohort

studies." I would say "it provides powerful filtering and

reporting options". I find "make sense" to be too informal for a

scientific paper.

5. The Author summary contains several typos also some have

grammatical mistakes as well.

6. Lines 34-35: "but does not take into account the fact that the

variants belongs in a cohort." I don't know what this sentence

refers to exactly. Also, I have the same comment for line 52: "any

chort characteristic". I think there should be a clear discussion

on what these are and how they are used or not used by the

different software tools.

7. ANNOVAR interface and annotated variant file: lines 66-69. This

section contains grammatical mistakes and is not clearly

structured. I think it would be good to have workflow chart to make

clear how different inputs can be used. Starting from VCF either

VCF --(3rd party tools) annoteted VCF or

VCF --(TAPES as a wrapper for ANNOVAR) annoteted VCF. And how to proceed

with the annotated VCF. I could not use the sort function on a VCF,

only on CSV.

8. Line 69: "without having to specify the databases and annotations

to use." It is true that when running one does not have to specify them,

but at set up the user has to specify which databases are to be

used, according to my experience. Also this comment gives the

impression that the user has no control over which databases are

being used.

9. Lines 76-80. I think it would be nice to have a description on how

each criterion was implemented as supplementary at least. Then the

Authors could say that most criteria were straight forward to

implement (see suppl.), but the others we solved in the following

way, and then explain them.

10. Line 95. assumptions "are" made.

11. Line 125. I would not use "most confidence", but "highest level of

confidence".

12. Lines 128-129. I would suggest to reformulate the first sentence

to make it clearer.

13. Line 134. What does "in the class" mean?

14. Line 169. "sheer number of" I find this a bit too informal.

15. Figure 1. and surrounding text is not well formulated. It is

difficult to interpret the difference between "TAPES proba" and

"TAPES ACMG". I have only realized what the difference was once I

opened the supplementary table and saw the last two columns. I

think this could be improved.

16. Lines 166-167. How did the Authors arrive at the threshold values (0.35 and 0.8) for

probability scores? Was it to maximize the score on the

example/training dataset? I would suggest to use more than one dataset

for benchmarking and testing. It should be avoided to optimize a

method on the benchmarking set.

17. Figure 1. According to the benchmarks TAPES falls either between the two other

software or performs worse than the other two software if we use the

ACMG results according to the ROC and precision recall analysis. While

this is not discussed in the text. Also would the Authors suggest to

use the Probability instead of ACMG then?

18. Figure 3. I prefer graphs with two axes. Having two y-axes makes

it difficult to interpret. The two graphs could be shown below each

other (A and B) with the same x-axis, but separate y-axes for the two plots.

19. Lines 188-194. I think this section could be improved by adding

example output, adding context on how does it compare to a

workflow without TAPES to fully show the benefits of the method.

I suggest to separate real data and mock up (made up) data

examples.

20. TAPES is able to assign variants to ACMG categories and then can

do further sorting and reporting. Other software tools can also

use ACMG categories as mentioned in the introduction. Can TAPES

use the output of those software and then do sorting and reporting?

21. On which platforms was TAPES tested?

22. Please add a release for TAPES that is referred to in the article.

Also add version numbers for the software used (or commit tags from GitHub).

## Summary

Overall, I find the tool promising. However, I do think both the

software package and the article require significant revision.

I do believe that TAPES can become a valuable tool.</module>

**Have all data underlying the figures and results presented in the manuscript been provided?**

Reviewer #1: Yes

Reviewer #2: No: I could not identify the input data used for the benchmark.

PLOS authors have the option to publish the peer review history of their article (what does this mean?). If published, this will include your full peer review and any attached files.

Reviewer #1: Yes: Nathan Dunn

Reviewer #2: No

---

## [Decision Letter · Decision Letter 1]

7 Sep 2019

Dear Dr Xavier,

Thank you very much for submitting your manuscript 'TAPES: a tool for assessment and prioritisation in exome studies' for review by PLOS Computational Biology. Your manuscript has been fully evaluated by the PLOS Computational Biology editorial team and in this case also by independent peer reviewers. The reviewers appreciated the attention to an important problem, but raised some substantial concerns about the manuscript as it currently stands. At this time we are not willing to consider a revised manuscript unless you can provide the following information, in addition to adequately answering the reviewers' concerns:

- the input file for the benchmark

- the reference set for the benchmark

- how the thresholds were calculated.

We cannot, of course, promise publication, even if you decide to send us a revised version.

Sincerely,

Mihaela Pertea

Software Editor

PLOS Computational Biology

Mihaela Pertea

Software Editor

PLOS Computational Biology

[LINK]

Reviewer's Responses to Questions

**Comments to the Authors:**

Reviewer #1: Thanks you for addressing my concerns.

Reviewer #2: In my opinion the manuscript is clearer now thanks to the corrections. I still think that adding a flowchart on input, output and processes could be very helpful to easily understand the pipeline and the possibilities. Below I have included a suggested workflow chart.

Most importantly, please indicate clearly the input file and the reference set for the benchmark, and how the thresholds were calculated. Otherwise, readers cannot evaluate the validity, it would be only faith or distrust.

In addition, I recommend to further improve the git repo, because potential users will give up easily if it is not clear or there are two manny mistakes. Adding a tutotorial with example input (e.g. the benchmark would be an excelent example), commands to run and the interpretation would help users test that everything is installed correctly and help them understand how the program works and what the seteps are.

Example flowchart, based on the manual, in mermaid (it can be drawn using the online editor: https://mermaidjs.github.io/mermaid-live-editor/):

```

graph TD

VCF an{annotate}

VCF an2

subgraph ANNOVAR

an2{annotate: table_annovar.pl}

end

an2 VCF2[VCF: annotated variants]

an2 TSV[TSV: annotated variants]

VCF2an

TSV |?|sort

subgraph TAPES

subgraph wrapped ANNOVAR

an

end

an  Annot[CSV: annotated variants]

Annot sort{sort}

sort Sorted[CSV: sorted varainats]

Sorted X{analyse}

end

X |by_sample| S[By sample report]

X |by_gene| G[By gene reoprt]

X |enrich| E[EnrichR report]

X |list| L[Kegg, List and Disease reoprt]

```

# Questions based on the response from the Authors

In one of the answers the Authors mention that for the benchmarking they used the dataset from the CharGer publication. Please include this also in the manuscript, and also add to the repository. The same answer ends with the comment that a sentence has been added to line 176. I think that line numbering has changed, so I could not find the referenced sentence.

The Authors claim that ANNOVAR wrapping is totally optional, although I could not run any of the commands without setting up the database by first installing ANNOVAR.

I still don't understand how the threshold values 0.35 and 0.8 were chosen for the probability method which is the recommended method. My assumption is that based on the benchmark set the Authors identified which cutoffs would yield the maximum number of correctly identified variants. If this is true then an independent data set is needed to test how valid the calls are, because the benchmark and training set should be independent from each other. If this assumption is false, please include the method used for deciding the threshold values.

# Textual comments

Line 35: "does not take into account the abundance of a variants in a cohort" should be "do not take into account the abundance of variants in a cohort"

Line 66-70. I could not perform the described steps without installing ANNOVAR and the setting up the database. Either include clearly in the manual how this can be done or modify this paragraph.

Line 116 should read "TAPES provides an array of different useful reports."

Line 119. "on the command line" not "in"

Line 121. "a pathway" not plurar

Line 122. "users do research" could be "run searches"

Line 145. TAPES will or does?

Line 158. Is the table only used as reference or also as input for the analysis? Please add the input.

Line 167-168. How do the ROC curves suggest the threshold values?

Figure 3: Why is the old TAPES curve used instead of the new one that is already in the git repo?

# Code review

The code still contains bugs that cause it to crash. Although the git repo suggests using `python tapes.py`, since tapes.py is written in python3 and the default python on many linux systems is python2 the program crashes.

Many of the example commands contains incorrect hyphen character that results in an error when copy pasting them to command line.

Attempting o run the "Quick Start" section

`python tapes.py db -s -A /path/to/annovar/` -> `python3 tapes.py db -s -A ~/temp/tapes/annovar/`

Worked fine with absolute path, but fails with relative path with a non informative error.

`python3 tapes.py db -s -A ../tapes/annovar/` Gives the following output:

```

No acmg_db path given and no db_config.json found

Default is: /home/user/temp/tapes-0.1/acmg_db

***TAPES: SEE DATABASE***

2019-09-04 13:45:59.....Fetching ANNOVAR Alldb file

NOTICE: Web-based checking to see whether ANNOVAR new version is available ... Done

NOTICE: Downloading annotation database http://www.openbioinformatics.org/annovar/download/hg19_avdblist.txt.gz ... OK

NOTICE: Uncompressing downloaded files

NOTICE: Finished downloading annotation files for hg19 build version, with files saved at the '.' directory

NOTICE: Web-based checking to see whether ANNOVAR new version is available ... Done

NOTICE: Downloading annotation database http://www.openbioinformatics.org/annovar/download/hg38_avdblist.txt.gz ... OK

NOTICE: Uncompressing downloaded files

NOTICE: Finished downloading annotation files for hg38 build version, with files saved at the '.' directory

Traceback (most recent call last):

File "tapes.py", line 406, in <module>

tf.check_online_annovar_dbs(annovar_path)

File "/home/user/temp/tapes-0.1/src/t_func.py", line 883, in check_online_annovar_dbs

with open(outfile_hg19, 'r') as file:

FileNotFoundError: [Errno 2] No such file or directory: '../tapes/annovar/hg19_avdblist.txt'

```

`python tapes.py db -b --acmg --assembly hg19` -> `python3 tapes.py db -b --acmg --assembly hg19` Fails

```

No acmg_db path given and no db_config.json found

Default is: /home/user/temp/tapes-0.1/acmg_db

***TAPES: DOWNLOAD DATABASE***

No annovar path given and no db_config.json found

Traceback (most recent call last):

File "tapes.py", line 358, in <module>

tf.build_annovar_db(annovar_path, args.assembly, args.acmg)

NameError: name 'annovar_path' is not defined

```

`python3 tapes.py annotate -i toy_dataset/toy.vcf -o test/output.vcf --acmg –a hg19` does not run and prints out the help page plus the following warning:

```

tapes: error: unrecognized arguments: –a hg19

```

After changing the hyphen to the correct character ` python3 tapes.py annotate -i toy_dataset/toy.vcf -o test/output.vcf --acmg -a hg19`

```

No acmg_db path given and no db_config.json found

Default is: /home/user/temp/tapes-0.1/acmg_db

***TAPES: ANNOTATE***

No annovar path given and no db_config.json found

Traceback (most recent call last):

File "tapes.py", line 384, in <module>

tf.process_annotate_vcf(args.input, args.output, annovar_path, args.assembly, args.ref_anno, args.acmg)

NameError: name 'annovar_path' is not defined

```

`python3 tapes.py sort -i toy_dataset/toy_annovar_multi.vcf -o test-sort/ --tab` works and creates a folder with three plots (png) and `test-sort.txt`, which is a tab separated file

`python3 tapes.py analyse -i test-sort/test-sort.txt -o test-report/report.txt --single_option` Fails with the following error:

```

tapes: error: unrecognized arguments: --single_option

`python3 tapes.py analyse -i test-sort/test-sort.txt -o test-report/report.txt` Runs without error, but creates no output

```

No acmg_db path given and no db_config.json found

Default is: /home/user/temp/tapes-0.1/acmg_db

***TAPES: RE-ANALYSE***

2019-09-04 14:07:10.....48 samples found

2019-09-04 14:07:10.....Output type: TXT/TSV + XLSX

2019-09-04 14:07:10.....Done

```

However, `python3 tapes.py sort -i toy_dataset/toy_annovar_multi.vcf -o test-full/ --tab --by_gene --by_sample --enrichr --list "MLH1 MSH6 MSH2" --disease "autosomal dominant" --kegg "pathways in cancer"` does work.</module></module></module>

**Have all data underlying the figures and results presented in the manuscript been provided?**

Reviewer #1: Yes

Reviewer #2: No: Input or reference set for the bechmarking or their clear description

PLOS authors have the option to publish the peer review history of their article (what does this mean?). If published, this will include your full peer review and any attached files.

Reviewer #1: Yes: Nathan Dunn

Reviewer #2: No

---

## [Decision Letter · Decision Letter 2]

1 Oct 2019

Dear Dr Xavier,

We are pleased to inform you that your manuscript 'TAPES: a tool for assessment and prioritisation in exome studies' has been provisionally accepted for publication in PLOS Computational Biology.

In the meantime, please log into Editorial Manager at https://www.editorialmanager.com/pcompbiol/, click the "Update My Information" link at the top of the page, and update your user information to ensure an efficient production and billing process.

One of the goals of PLOS is to make science accessible to educators and the public. PLOS staff issue occasional press releases and make early versions of PLOS Computational Biology articles available to science writers and journalists. PLOS staff also collaborate with Communication and Public Information Offices and would be happy to work with the relevant people at your institution or funding agency. If your institution or funding agency is interested in promoting your findings, please ask them to coordinate their releases with PLOS (contact ploscompbiol@plos.org).

Thank you again for supporting Open Access publishing. We look forward to publishing your paper in PLOS Computational Biology.

Sincerely,

Mihaela Pertea

Software Editor

PLOS Computational Biology

Mihaela Pertea

Software Editor

PLOS Computational Biology

Reviewer's Responses to Questions

**Comments to the Authors:**

Reviewer #2: Dear Authors,

Thank you for addressing all my comments. I think the manuscript has improved significantly since the submission. I find the new comparison results and figures very impressive and convincing.

I have two minor comments:

Is the release number still 0.1 as stated in the manuscript or is it 0.1.1? I would suggest the improved version. Otherwise, potential users might start with 0.1 and be discouraged by the bugs and give up.

I would include the version of Figure 3 that best represents the version of the software that is used for the latest version of the manuscript and github.

I leave both these comments up the the Authors consideration when working on the final proof of the paper.

**Have all data underlying the figures and results presented in the manuscript been provided?**

Reviewer #2: Yes

PLOS authors have the option to publish the peer review history of their article (what does this mean?). If published, this will include your full peer review and any attached files.

Reviewer #2: No

---

## [Editor Report · Acceptance letter]

9 Oct 2019

PCOMPBIOL-D-19-01091R2 

TAPES: a tool for assessment and prioritisation in exome studies

Dear Dr Xavier,

I am pleased to inform you that your manuscript has been formally accepted for publication in PLOS Computational Biology. Your manuscript is now with our production department and you will be notified of the publication date in due course.

With kind regards,

Matt Lyles
